# Leisure-Time Activities in Different Contexts and Depressive Symptoms in Norwegian Adolescents: A Cross-Sectional Study

**DOI:** 10.3390/ijerph191710769

**Published:** 2022-08-29

**Authors:** Annette Løvheim Kleppang, Eivind Å. Skille

**Affiliations:** Department of Public Health and Sport Sciences, Inland Norway University of Applied Sciences, 2418 Elverum, Norway

**Keywords:** adolescents, physical activity, leisure-time activity, capital forms, depressive symptoms

## Abstract

The purpose of this study was to examine the association between physical activity organised in sports clubs, non-organised physical activity, other organised leisure-time activities, and depressive symptoms among adolescents. This study was based on cross-sectional data from the Ungdata survey conducted between 2017 and 2019. The sample comprised 7656 adolescents (aged 13–16 years). Binominal logistic regression was used to analyse the association between the three different leisure-time activities and depressive symptoms. All models were adjusted for gender, family economy, parents’ higher education, having friends, alcohol intoxication, and smoking. The odds for symptoms of depression were higher for those who were less physically active in a sports club (OR: 1.34, 95% CI: 1.15–1.57) and in non-organised physical activities (OR: 1.50, 95% CI: 1.29–1.74) and lower for those who participated less in other organised leisure-time activities (OR: 0.79, 95% CI: 0.68–0.92) compared with those who were physically active (sports club and non-organised) and those who participated in other organised leisure-time activities. Our findings suggested that being physically active, both in a sports club and in non-organised activities, was associated with lower odds of depressive symptoms among adolescents. Additional research is needed to confirm a possible causal relationship.

## 1. Introduction

### 1.1. Depressive Symptoms among Adolescents and Possible Mechanisms

Mental health problems are considered a significant challenge for both individuals and societies. Worldwide, depression is a common mental disorder that is characterised by persistent sadness and a lack of pleasure or interest in previously rewarding or enjoyable activities [1]; depression is the fourth-leading cause of disability and illness among adolescents [2]. During adolescence, social cognition undergoes profound development [3] as the impact of puberty on the brain makes adolescents particularly sensitive to their social environment [4]. Moreover, approximately half of all known mental health issues seem to originate from youth, although this is more evident among girls than boys [5]. Mental health problems, such as depressive symptoms, can be considered phenomena that most, if not all, human beings have more or less experienced. In the words of Stevens, Lieschke, Cruwys, Cárdenas, Platow, and Reynolds [6], ‘depression is best viewed as a continuum on which everybody falls’ (p. 1). A systematic review among adolescents concluded that the global point prevalence rate of depressive symptoms increased from 24% between 2001 and 2010 to 37% between 2011 and 2020 [7]. Additionally, results from previous studies showed that depressive symptoms have increased, especially among girls [8,9]. Therefore, identifying factors that can reduce the risk of depressive symptoms is important.

Physical activity has been characterised as an important and positive behaviour in promoting mental health in children and adolescents [10,11]. Additionally, there seems to be a relatively overarching consensus that physical activity can be one possible mechanism for protection against depression [12]; in this respect, there are two significant and more specific arguments supporting this general statement. First, there is a vast amount of evidence suggesting that an increased frequency of participation in physical activities leads to fewer depressive symptoms and better well-being [13,14,15]. For example, Dishman, McDowell, and Herring [13] identified that, among adults, the odds of developing depressive symptoms lowered significantly if public authority guidelines were followed and one was moderately physically active on a stable basis (compared to on a single-event basis) and that the odds were lowered if this was maintained over time, while Schuch and Stubbs [15] emphasised the necessity to stay in (not drop out of) physical activity programs to gain the effects of prevention against depression. Second, there seems to be an inter-human or social element intertwined with the argument of physical activity for reduced depression. In this respect, Bone, Bu, Fluharty, Paul, Sonke, and Fancourt [16] proposed that leisure-time activities among adults seem to have potential protective functions against depression, among other reasons, due to the social interactionist nature of the activities. They reported positive outcomes (against depression) regarding activities that could be defined as both ‘being social’ and just offering people ‘something to do’ [16]. Graupensperger, Panza, Budziszewski, and Evans [14] claimed that social identification within groups, especially when developed over some time, can contribute to wellbeing. This includes and can relate to family issues, which we return to below.

Here, before we move on to our outset, we position ourselves and our study in line with Stevens, Lieschke, Cruwys, Cárdenas, Platow, and Reynolds [6], who pinpointed how the research before their study about the positive psychological outcomes of adults participating in physical activities that include group memberships of some sorts did not examine ‘whether depression-related benefits might arise from *belonging to groups* that engage in physical activity’ (p. 1, original emphasis). We developed this research branch by focusing on how various leisure contexts in adolescence regarding physical activity and sociality are associated with depressive symptoms. Moreover, leisure activities extend into a more general unity of lifestyle habits, including health-related ones. In this respect, we highlight two additional empirical elements and one analytical, theoretical point from Ohrnberger, Fichera, and Sutton [17]. The empirical aspects are that there are associations between alcohol consumption and depression and smoking and depression [18,19,20,21]. The analytical point we copy from Ohrnberger, Fichera, and Sutton [17] is the application of Bourdieu’s theory, especially the focus on various capital forms, to analyse lifestyle habits more generally, including the use of alcohol and tobacco, and leisure-time activities and depression more specifically. With such a sociological approach, an integrated interest in family socialisation is unavoidable.

### 1.2. The Role of the Family in Physical Activity and Leisure Habits

This study had two aims and hence contributes to two research fields. On the one hand, we lean on and contribute to the field of research on physical activity and mental health problems. More specifically, and as per above, within a rather vast amount of research, we connect with research examining the relationship between physical activity and depressive symptoms. As far as we are concerned, there is a consensus regarding physical activity and depression, which reveals a clear negative relationship suggesting that physical activity is a positive aid in mental health problems. However, physical activity is not a single and united entity; it can take many forms, both activity-wise and not least regarding its organisation. In other words, the well-known and empirically established knowledge about the negative correlation between physical activity and symptoms of depression are more fine-grained than often believed and must be nuanced in its research.

In this respect, Kleppang, Hartz, Thurston, and Hagquist [22] showed that youth participating in sports clubs or other forms of organised physical activity had lower odds for symptoms of depression (compared with youth physically active on their own or in a gym, or those that are physically passive). This suggests that there might be something with the context of social interaction that leads to fewer depressive symptoms than (only) physical activity per se.

Given this outcome, the other research field that this study stemmed from and contributes to is sociological studies of participation in organised sports. More precisely, we were influenced and inspired by studies that explored patterns of participation and explain such patterns with underlying variables. For example, Strandbu, Bakken, and Sletten [23] found that participation in organised sports—which, in Norway, usually refers to sports clubs and includes both girls and boys [24]—is, to a large degree, explained by ethnicity and family background, as well as class [25].

The relatively vague term ‘family background’ comprises both classical sociological dimensions, such as economic status (often measured by household income or perception of being financially fine), parents’ education and/or occupation, and the family’s cultural habits. Moreover, Strandbu, Bakken, and Stefansen [26] found that a family’s sporting habits, to a large degree, determine the participation possibilities for youth. In this respect, based on general sociological interest and a theoretical approach to which we soon return, we believe that classical sociological explanations can contribute to the analysis of both depressive symptoms and participation in leisure activities and the analysis of the associations between the two.

### 1.3. Aim and Research Questions

Taken together, we found ourselves and this study placed across research fields. In this study, we merged these two outlets to investigate in more detail whether it is the physical activity element and/or the social, organisational element of sports and other leisure-time activities that can be considered potential mechanisms to prevent depressive symptoms. The aim of this study was thus to examine the associations between family background, leisure activities in three various contexts (including sports and physical activity), and depressive symptoms. In this study, we aimed to answer the following research question and sub-questions:

What are the associations between organised physical activity in sports clubs, non-organised physical activity, other organised leisure-time activities, and depressive symptoms among Norwegian adolescents?

How do the capital forms (economic, cultural, and social) relate?How do other health variables, such as alcohol intoxication and tobacco, relate?Are the strengths of the association between leisure-time activities and depressive symptoms influenced by gender and family economy?

We employed a theoretical conceptualisation offered by Bourdieu, which involved a set of concepts that can assist in analysing sports participation. Moreover, as long as there are examples of Bourdieu applied to mental health studies of youth [27,28], we find Bourdieu apt for this study. More exactly, we applied Bourdieu’s notion of capital, namely, economic, cultural, and social aspects (elaborated in the Theory section below), because it was shown to be applicable in similar studies and because it is possible to operationalise given the data that we have available (see the Methods section for an outline of the data employed).

### 1.4. Theory

In the study of family background, organised leisure, and symptoms of depression, we were inspired by the so-called health lifestyles theory (HLT) that could assist in analysing how social structure and individual agency interplay to influence health-related practices [29,30] by combining Weber’s [31] conceptions of how life changes (structure) and life choices (agency) interplay and Bourdieu’s [32] theory about how the amount and composition of various capital forms influence lifestyle practices [29,30]. Without denying the applicability of HLT, we considered the Weberian perspective as part of the Bourdieuan perspective; thus, we applied Bourdieu’s theory, which is rooted in a relational and multidimensional orientation to explain causality [33] and considers the homological relationship between social class and practices because it examines how individuals occupy specific positions in a multidimensional social space that constitutes membership in a particular social class. In other words, Bourdieu’s theory considers habitus.

Habitus refers to a system of dispositions that guides people’s choices and attitudes [32], which expresses itself in all domains of life, including preferences for leisure-time and health-related practices. Habitus develops throughout life, and individuals are predisposed to certain choices because they are exposed to practices and habits by their immediate contexts, including people living in similar life conditions [32]. In other words, individuals, particularly youth, are socialised by those close to them, most prominently their family. Consequently, people with similar positions in social space and youth with parents with similar characteristics regarding education, occupation, income, and leisure-time practices tend to develop similar habiti, and consequently, similar lifestyles. Hence, there is a homology between the social space and the choice of lifestyles [32]. While habitus is a rather abstract concept, to operationalise the positioning of individuals in social space, Bourdieu proposed the concept of capital [32], which is based on the metaphor of—but also criticised the focus on—monetary exchange when he defined capital as ‘accumulated labour (in its materialised form or its “incorporated” embodied form)’ (p. 241).

Bourdieu proposed three main forms of capital, namely, economic, cultural, and social, which can be applied to define an individual’s position relative to others in social space; the individual is positioned by the amount of capital that they possess and by the relative distribution of different types of capital (i.e., how much of each form). Economic capital refers to classic economic assets: income and fortune—or money, house, cars, etc. Cultural capital refers to general knowledge and to the ability to understand more specific codes. Cultural capital can take three sub-forms: institutionalised cultural capital, which refers to formal education; embodied cultural participation; and objectified cultural capital, which refers to the possession of cultural goods. Social capital refers to resources a person has or can gain through a social network [32]. Thus, persons that possess greater amounts of economic, cultural, and social capital are more privileged, whereas habitus is shaped by measurable capital forms. We operationalised these definitions with the variables discussed in the Methods section. Although Bourdieu never conducted research on health himself, his concepts of capital have proven apt for the study of health inequalities because each capital form and the interplay between them can be considered relevant to explain health practices [27,28].

## 2. Materials and Methods

This study was based on data from the Ungdata Survey, which is an ongoing national survey of adolescents across all municipalities in Norway that has become an essential source of information on young people’s health and wellbeing, both at the national and municipal levels [34]. The survey has been conducted by the Norwegian Social Research (NOVA) at Oslo Metropolitan University in cooperation with the Regional Drug and Alcohol Competence Centres (KoRus Øst). The surveys are partially financed by the Norwegian Directorate of Health and cover different aspects of adolescents’ lives, i.e., health issues and school issues; relationships with friends and parents; leisure-time activities, including participation in physical activities and organised sports; dietary habits; and symptoms of depression. It also includes questions about alcohol consumption and tobacco use.

In this study, we included adolescents from lower secondary schools (aged 13–16 years) from Innlandet county who answered between 2017 and 2019. The adolescents took approximately 30–45 min to complete the questionnaire. Altogether, 7656 adolescents from 29 municipalities participated, giving an overall participation rate of 87% in lower secondary schools at a national level [35].

The web-based questionnaire was administered at school during school hours with an administrator or a teacher present to answer questions that arose. The parents were given information regarding the survey and had the opportunity to withdraw their children from participation in advance. Additionally, the adolescents were informed that participation was voluntary. The data were analysed by independent researchers who did not participate in the data collection. The study was conducted in line with the Declaration of Helsinki, and the Norwegian Centre for Research Data (NSD) approved all ethical aspects of the study.

### 2.1. Measures

#### 2.1.1. Symptoms of Depression

The six items measuring depressive symptoms were derived from a scale that was based on the Hopkins Symptom Checklist—90 [36]). The adolescents were asked whether they had been affected by any of the following during the past week: ‘Felt that everything is a struggle (item 1)’; ‘had sleep problems (item 2)’; ‘felt unhappy, sad or depressed (item 3)’; ‘felt hopelessness about the future (item 4)’; ‘felt stiff or tense (item 5)’; and ‘worried too much about things (item 6)’. The six questions had four response categories: ‘Not been affected at all (1)’, ‘not been affected much (2)’, ‘been affected quite a lot (3)’, and ‘been affected a great deal (4)’. Mean scores were computed, ranging from 1 to 4; high scores indicated higher symptoms of depression. In the present study, the depressive symptom scale was dichotomised based on a cut-off point of the 80th percentile (2.6667), producing two categories on the scale: high level of depressive symptoms (≥80th percentile) and low level of depressive symptoms (<80th percentile). The depressive scale was psychometrically evaluated among Norwegian adolescents. The scale showed good reliability (Person separation index: 0.802), the targeting was acceptable, the response categories were ordered, and as a whole, the scale worked reasonably well at a general level [37].

#### 2.1.2. Leisure-Time Activities in Different Contexts

Physical activity in sports clubs: The participants were asked how often they participated in the following activity categorised as ‘exercising or competing with a sports club’. The response alternatives included ‘never’, ‘rarely’, ‘1–2 times a month’, ‘1–2 time a week’, ‘3–4 times a week’, and ‘at least 5 times a week’. Participation was dichotomised into ‘≤1–2 times a week’ and ‘>1 time a week’.

Non-organised physical activities: Non-organised physical activity was assessed by asking: ‘Below we have listed various activities that you can do in your free time. Think about the last week (last 7 days). How many times have you Played football, snowboarded or did another type of physical activity together with friends (not with a sports club)?’ The response alternatives included ‘no times’, ‘once’, ‘2–5 times’, and ‘6 times or more’. Participation was dichotomised into ‘≤1 time a week’ and ‘>1 time a week’.

Other organised leisure-time activities: The participants were asked how many times during the last month they had participated in activities, meetings, or training organised by the following associations, clubs, or societies, which were categorised as follows: ‘after-school club/youth centre/youth club’; ‘religious organisation’; ‘band, choir, orchestra’; ‘culture school/music school’; and ‘other organisation, society or associations’. Response alternatives included ‘never’, ‘1–2 times’, ‘3–4 times’, and ‘5 times or more’. Participation was dichotomised into ‘≤3–4 times a month’ and ‘>1–2 times a month’, and then each variable was computed to provide a total variable that measured whether the participants had participated in at least one of the organised activities.

Economic capital: The family economy was measured as follows: ‘Financially, has your family been well off, or badly off, over the past years?’ The response alternatives were as follows: ‘we have been well off’, ‘we have neither been well off nor badly off’, ‘we have generally been badly off’, and ‘we have been badly off the whole time’. The family economy was recoded as ‘good’, ‘neither bad nor good’, and ‘bad economy’ (see Table 1).

Cultural capital: Parents’ level of education was measured separately for each parent by asking the following: ‘Did your father/mother go to university or to a university college?’ Those young people not in touch with one or both parents were asked to leave the answer blank. This was categorised as ‘yes’ or ‘no’. Parental educational status was stratified as ‘no university education’, ‘one parent with university education’, or ‘both parents have with university education’.

Social capital: Friendships were measured by asking: ‘Do you have at least one friend who you trust completely and who you can tell absolutely anything?’ This was categorised as ‘Yes, definitely’, ‘Yes, I think so’, ‘I don’t think so’, and ‘I have nobody I would call close online friends at the moment’. This was categorised as ‘Yes, I have friends’ and ‘No, I do not have friends’ in the present study.

Other (un)healthy habits: Alcohol intoxication was based on how many times adolescents reported having been intoxicated during the past 12 months. The response alternatives were as follows: ‘never’, ‘once’, ‘2–5 times’, ‘6–10 times’, and ‘more than 11 times’. Intoxication was dichotomised as ‘no intoxication episodes’ and ‘any intoxication episode’.

Smoking was assessed by asking: ‘do you smoke?’ The response alternatives were ‘I’ve never smoked’, ‘I used to smoke, but I’ve stopped completely now’, ‘I smoke less than once a week’, ‘I smoke every week, but not every day’, and ‘I smoke every day’. Smoking was dichotomised as ‘no current use’ and ‘current use’.

Table 1 gives the Ungdata Survey 2017–2019 questions, response alternatives, and variable definitions included in this study.

### 2.2. Statistical Analysis

Analyses were carried out using SPSS 26.0 (IBM Corp, Armonk, NY, USA) for Windows. Baseline characteristics stratified by depressive symptoms and gender are presented as proportions with a 95% confidence interval (CI) in each stratum. No overlap of the CI was considered significant at the 5% level.

Binomial logistic regression analyses was performed to examine the association between leisure-time context and depressive symptoms. All three models were adjusted for gender, family economy, parents’ education, having friends, alcohol intoxication, and smoking habits. As the variable of other ‘organised leisure’ activities originally comprised many specific activities (cf. the Measures section: ‘after-school club/youth center/youth club’; ‘religious organization’; ‘band, choir, orchestra’; ‘culture school/music school’; and ‘other organization, society or associations’), separate analyses were completed for each, while the results showed the same pattern as a collapsed variable; therefore, only the latter is reported. Associations are presented as odds ratios (ORs) with 95% confidence intervals (CIs).

Interaction analysis was used to examine the influence of gender and family economy on the strength of the relationship between physical activity in a sports club and depressive symptoms. Possible interaction effects were examined using LR tests (likelihood ratio tests) by contrasting models with and without interaction terms. The main effect model included physical activity in a sports club, gender, family economy, having friends, alcohol intoxication, and smoking as independent variables and was tested against models that included interactions between physical activity in a sports club by gender, physical activity in a sports club by family economy, physical activity in a sports club by having friends, physical activity in a sports club by smoking, and physical activity in a sports club by alcohol intoxication. Parallel interaction analysis was carried out for non-organised physical activity and other leisure-time activities. Only one interaction effect was significant (physical activity in a sports club by family economy), and only this is reported.

## 3. Results

Table 2 shows the baseline characteristics of the study population aged 13–16 years in 2017–2019 according to depressive symptoms stratified by gender.

Table 2 shows that a significantly higher proportion of girls reported a high level of depressive symptoms compared with boys (23.8% vs. 8.3%). Overall, and in gender subgroups, those with a high level of depressive symptoms reported a significantly poorer family economy than the rest of the study population. Among youth with a low level of depressive symptoms, a higher proportion participated in a sports club one or more times per week, a higher proportion participated in non-organised physical activity one or more times in the last seven days, and a lower proportion spent three hours or more in the last month in other leisure-time activities compared with those with less physical activity and those with a higher level of other leisure-time activities. Furthermore, a higher proportion of the youth with lower levels of depression symptoms reported having friends and parents with a higher level of education. Additionally, youth with lower levels of depressive symptoms reported no alcohol-intoxicating episodes more often during the previous 12 months and no tobacco use than the rest of the population. These patterns were observed in boys and girls separately.

In Table 3, we see that, overall, 12.4% of the adolescents who participated in a sports club or non-organised physical activity reported depressive symptoms. In other organised activities, the proportion of those with depressive symptoms was 18.2%. A higher proportion of males participated in non-organised physical activities, and a lower proportion participated in other leisure-time activities compared with females. Approximately 64% of the adolescents attending a sports club had two parents with higher education, similar to those participating in non-organised PA activities. In other organised activities, the proportion was 58.2%. Among adolescents who went to a sports club, 2.3% smoked and 9.2% reported alcohol intoxication in the last 12 months. In other activity contexts, the proportion varied between 11.0 and 12.7% for alcohol intoxication and 3.3 and 3.7% for smoking.

Based on the descriptive statistics presented in Table 3, we conducted a binary logistic regression analysis with the dichotomised depressive symptoms scale as the dependent variable, the results of which are shown in Table 4.

In the crude model, the odds of having symptoms of depression were higher for those who were less physically active in a sports club (OR: 1.70, 95% CI: 1.50–1.93), higher for those taking part in non-organised physical activities (OR: 1.87, 95% CI: 1.65–2.11), and lower for those who participated less in other organised leisure-time activities (OR: 0.76, 95% CI: 0.67–0.86) compared with those who were physically active and those who participated in other organised activities. After adjusting for other independent variables, only small changes in the OR were observed.

Across all models, the odds of having symptoms of depression were higher in girls than in boys (OR: 3.44–3.68) and higher for those who reported not having friends (OR: 2.48–2.80), having a poor economy (3.28–3.60), smoking (2.68–2.82), and any intoxication episodes (OR: 2.14–2.24) compared with those who reported having friends, a good economy, not smoking, and not having an intoxication episode, respectively. The likelihood ratio test showed that the model with the interaction term ‘physical activity in a sports club by family economy’ fit the data significantly better than the multivariate main effects model. This interaction implied that perceived family economy significantly modified the association between physical activity in a sports club and depressive symptoms. Following the findings of the above regression analysis, we conducted a separate regression analysis for those with good, neither good nor bad, and poor family economies, which is presented in Table 5.

The results of the moderation analysis after an adjustment for confounders are presented in Table 5. In general, family economy demonstrated a moderated effect on the relationship between low physical activity in a sports club and symptoms of depression. Among adolescents with a good family economy, participating in a sports club fewer than one time a week led to a 1.56 times greater likelihood of depressive symptoms than those who participated one time or more per week. Considering those with a poor family economy participating in a sports club fewer than one time a week, the results became non-significant.

## 4. Discussion

The results from the present study revealed an association between a lower level of physical activity (both in sports clubs and non-organised activities) and a higher level of symptoms of depression. Further, our results confirmed an association between those who participated less in other organised leisure-time activities and lower levels of symptoms of depression. Moreover, an interesting detail in our findings was that the association between physical activity in a sports club and depressive symptoms differed between adolescents reporting good, neither good nor bad, and poor family economies. Thus, we discuss whether family background can be part of the explanation of the association between leisure-time contexts and depressive symptoms and other (un)healthy phenomena, such as smoking and alcohol use. In this respect, there are a couple of details from the Results section that we want to discuss in more detail: first, which leisure-time contexts were associated with higher and lower depressive symptoms, and second, how the capital forms were potential explanatory mechanisms for the pattern.

Our findings correspond with a meta-analysis of prospective cohort studies indicating that a higher level of physical activity has a protective effect on future depression [12] and a review of reviews indicating a causal association between physical activity and depression [10]. However, Toseeb et al. [38] reported no association between physical activity and symptoms of depression. Thus, physical activity and mental health problems are complex measures and need to be operationalised as such in research examining the association between the two. How mental health problems are measured and operationalised may have an impact on their relationships with physical activity.

We have to admit that when we started this study, we were conceptually based on findings from [22] suggesting that organised and thus social leisure activities would cover more explanatory value regarding reduced depressive symptoms compared with physical activities [39,40]. In this respect, we were a little bit surprised when our findings so clearly showed that physical activities, whether organised in sports clubs or played together with friends in an informal setting, showed fewer associations with symptoms of depression compared with activities that were ‘only’ social.

Moreover, dispositions that are developed through family socialisation, referred to by Bourdieu as habitus, seem to form patterns of both leisure-time participation, the risk for depressive symptoms, and not least, the relationship between socialisation-related variables and various health-related variables. While habitus refers to a subjective feeling—including taken-for-granted elements—about oneself (for example, the feeling of being economically poor), we operationalised the relatively abstract notion of habitus with the more concrete concept of capital; hence, we found that economic and social status have potential explanatory value regarding depressive symptoms among youth, while cultural capital, measured as the parents’ education, did not (cf. Table 4).

Cultural capital, measured as the parents’ education, did not explain much about the risk for depressive symptoms among Norwegian youth. This indicated a reason why cultural capital has received little attention in health research [41]; however, when cultural capital is theoretically compatible and exchangeable with other capital forms, and when we know how other forms of capital influence health, cultural capital should be part of an overarching discussion about health outcomes and inequalities [42,43]. Cultural capital could also be measured as tastes and values or as the possession of goods. Nevertheless, we checked for the number of books in the home, and it did not contribute to the explanation of the variation of depressive symptoms. We will not, however, dismiss the idea that embodied cultural capital can be interesting in interpreting health: ‘it is in this form that cultural capital becomes a key component that links people’s social position with the behavioural aspects of health inequality’ [41]. Neither do we deny that an element of cultural capital could be defined as health-related knowledge (health literacy), which we did not measure.

Regarding economic capital, which is understood here as material resources that are ‘immediately and directly convertible into money and may be institutionalised in the form of property rights’ [44] and measured by the youth’s perception of the family economy, we found a clear pattern of how the (perceived) amount of resources was positively related to health. Consequently, it could be interpreted that a poorer perceived family economy can influence the probability for an individual to report health problems, such as symptoms of depression. This rather subjective measure (perceived family economy) emphasises the psycho-social meaning of differences regarding economic capital (captured by the concept of habitus and its linkage to the objective, measurable concept of capital). Hence, perceiving less economic capital than others can probably cause feelings of powerlessness and unfavourably impact mental health. In this respect, our findings indicated that the association between low physical activity in a sports club and depressive symptoms differed between adolescents reporting a good, neither good nor bad, or poor family economy. That is to say, the strengths of the association between low physical activity in a sports club and depressive symptoms were affected by the family economy.

When it comes to social capital—which, for Bourdieu, is accrued to individuals as a resource of networks and interpersonal relationships (in Bourdieu’s words, ‘the aggregate of the actual or potential resources which are linked to the possession of a durable network of more or less institutionalised relationships of mutual acquaintance and recognition’ [44]), and in this study was measured by having friends—we found it had a potential explanatory value for lower depressive symptoms. Thus, in interpreting social capital’s impact on depression symptoms, we followed Song [45], who considered the term ‘social capital’ exclusively for the resources available in a social network and identified 10 mechanisms of how social capital can influence health. Such health-related mechanisms of social capital include reduced stress and isolation/loneliness, as well as healthier norms and habits that can transit through networks. In this respect, it is especially interesting that participation in less organised leisure activities (other than physical activities) gave a lower OR for depressive symptoms. After checking for each of these forms of activities (youth leisure club, religious association, music corps and choir, culture school, other association), the pattern remained.

Thus, our findings are in accordance with the established knowledge about physical activity being associated with lower degrees of depressive symptoms. This counted for both girls and boys, while the well-established pattern of gender difference was also identified in our study. To specify, being a girl was associated with a greater likelihood of having depressive symptoms.

Nevertheless, despite a rather opposite research population (Bone and colleagues studied older adults in the United States), we followed Bone, Bu, Fluharty, Paul, Sonke, and Fancourt [16], who stated that it is ‘difficult to disentangle whether the association between engagement and depression is due to self-selection, revere causality, or because these activities reduce depression’ (p. 6). In this respect, there seems to be a set of background variables that explain participation in leisure-time activities, depressive symptoms, and other health-related variables (exemplified here by alcohol and smoking habits).

In summary, the findings of his study may be considered as an overlap between various fields that suit people with a given habitus. Specifically, we can see the sketches of a field of physical activity that overlaps with the field of education and social life (middle-class youth who perceived a good family economy), in which people in these fields seemed to have lower symptoms of depression and lower levels of unwanted health behaviours. This is all in line with the theory of how an individual’s habitus corresponds to the individual’s position in social space, and that position in social space develops a disposition for certain habits and participation in particular fields [32]. In other words, this study of depressive symptoms among youth in Norway contributes to the social science debate about reflexive individualisation versus determination. Choices of, for example, leisure activities can, therefore, be measured if they are structured according to the variables posed in Bourdieu’s capital metaphor. Although some scholars claim that the new modernity is more or less classless (e.g., Giddens, 1991), there are a set of new classes that are created in the reflexive modernity; however, the most important point is, as Lash wrote three decades ago, that ‘the personnel filling these class positions are typically determined by more particularistic, “ascribed” characteristics’ (Lash, 1994, p. 134). The impact of socio-economic status was significant in explaining depressive symptoms. People end up in categories of ‘winners’ and ‘losers’ (Lash, 1994, p. 155); the classification into winners and losers depends on an individual’s habitus, which, in turn, is based on where in a social space they are clustered.

We find the theory of Bourdieu apt as a framework for interpreting differences in mental health problems because it provides a toolbox for not only describing them but for discussing and explaining differences in health inequalities by taking into account various resources that are possessed by individuals and patterned in society. It enables the inclusion of health indicators that are often neglected, hereunder the subjective perception of economic capital, as well as other cultural and social aspects. Most significantly, Bourdieu’s theory focuses not only on the resources that some people ‘lack’ but rather is ‘a theory of privilege than a theory of inadequacy’ (Pinxten and Lievens, 2014, p. 1097).

### Strength and Limitations

The data used in this study were collected in 2017–2019 and provide an up-to-date description of key aspects of adolescents’ lives. Additionally, the outcome measure worked well psychometrically at a general level [37]. A limitation of the study was the cross-sectional design, which precluded inferences about causal relationships. Depressive symptoms may function not only as an outcome but also may act as an exposure that hampers leisure-time activities and can influence how young people evaluate their leisure time. Additionally, the use of self-reported measures may have led to misclassification or measurement error. Furthermore, the associations between leisure-time activities and depressive symptoms might be influenced by other factors that were not controlled for in the present study.

## 5. Conclusions

In conclusion, we found that being physically active, both in a sports club and in non-organised activities, was associated with lower odds of depressive symptoms among adolescents in Norway. Moreover, we also found that the association between being physically active in a sports club and symptoms of depression was not significant for those reporting a poor family economy (cf. Table 5). This can be interpreted as stating that a poor family economy—or the perception of a poor family economy, to be specific—trumped the possible good of being physically active (in prevention terms) regarding depressive symptoms. In terms of the aims of this study, regarding the positioning of our study in line with Stevens, Lieschke, Cruwys, Cárdenas, Platow, and Reynolds [6] and their emphasis on how positive psychological outcomes of adults participating in physical activities probably had to include a feeling of group membership, we found, somewhat surprisingly, that among youth in Norway, physical activity of any level of organisation was fruitful from a mental health perspective. Having said that, it was not necessarily the case that non-organised physical activity equals being lonely; it might happen with friends.

When it comes to the political and practical implications of this research, the ‘depressive message’ is to support girls to participate more in physical activities and let people increase their economy or at least facilitate a better perception of the family economy among youth. These suggestions are impossible. Thus, to identify any causal relationships or mechanisms explaining these statistical relationships, more research is needed. For example, we suggest qualitative interviews of youth with a lower perceived family economy; in this respect, studies with a longitudinal design, and perhaps even better, a mixed-methods approach that covers both measured and perceived economies would be appropriate. Additionally, qualitative research could and should determine what non-physical activity really means for youth. If it refers to sports-like activities outside a formal organisation but in a social setting with friends, perhaps the fact that friends are involved, at least partially, may explain the association with lower odds of depressive symptoms. Exploring underlying mechanisms of how physical activity can improve mental health in both clinical and public health settings is vital to future research and health policies. Public health strategies should facilitate healthy lifestyle choices for all adolescents, independent of family support, e.g., through universal interventions, such as safe environments that promote physical activity.

## Figures and Tables

**Table 1 ijerph-19-10769-t001:** Ungdata Survey 2017–2019: questions, response alternatives, and variable definitions.

Questions	Response Alternatives	Variable Definitions
** *Symptoms of depression* **		
During the past week, have you been affected by any of the following issues:		
Felt that everything is a struggle (item 1)	Not been affected at all, not been affected much, been affected quite a lot, and been affected a great deal.	Symptoms of depression
Had sleep problems (item 2)		≥80th percentile
Felt unhappy, sad, or depressed (item 3)		
Felt hopelessness about the future (item 4)		
Felt stiff or tense (item 5)		
Worried too much about things (item 6)		
** *Physical activity* **		
How often do you participate in the following activities? Train or compete with a sports club	Never, rarely, 1–2 times a month, 1–2 times a week, 3–4 times a week, and at least 5 times a week.	<1 time a week, ≥1 time a week
Below we have listed various activities that you can do in your free time. Think about the last week (last 7 days). How many times have you?		
Played football, snowboarded, or performed another type of physical activity together with friends (not with a sports club)	Never, once, 2–5 times, 6 times, and more per week.	<1 time a week, ≥1 time a week
** *Other leisure-time activities* **		
How many times during the last month did you participate in activities, meetings, or training organised by the following associations, clubs, or societies.		
After-school club/youth centre/youth club		<3 times a month, ≥3 times a month
Religious organisation	Never, rarely, 1–2 times, 3–4 times, 5 or more times a month.	Total ≥3 times in at least one of the activities
Band, choir, orchestra		
Culture school/music school		
Other organisation, society, or association		
** *Friends* **		
Do you have at least one friend whom you trust completely and whom you can tell absolutely anything?	Yes definitely, yes, I think so, I do not think so, I have nobody I would call close online friends at the moment	Yes, no
** *Gender* **		
Are you a boy or a girl?	Boy, girl	
** *Smoking* **		
Do you smoke?	I have never smoked, I used to smoke but I have stopped completely now, I smoke less than once a week, I smoke every week but not every day, and I smoke every day.	No smoking, smoking
** *Alcohol intoxication (previous 12 months)* **	Never, once, 2–5 times, 6–10 times, more than 11 times	Any intoxication episode, no intoxication episodes
** *Parents higher education* **		
Did your father and mother go to university or to a university college? Select one answer for each parent. If you are not in touch with one or both of your parents, skip the question about that parent.	Yes, no	Both parents, one of the parents, neither of the parents
** *Family economy* **		
Financially, has your family been well off or badly off over the past few years?	We have been well off the whole time, we have generally been well off, we have neither been well off nor badly off, we have generally been badly off, we have been badly off the whole time	Good economy, neither bad nor good economy, bad economy

**Table 2 ijerph-19-10769-t002:** Youth data surveys: baseline characteristics of adolescents aged 13–16 years in 2017–2019 according to gender and depressive symptoms.

	Boys (n = 3802)		Girls (n = 3861)	
	Lower Level of Depressive Symptoms	Higher Level of Depressive Symptoms	Lower Level of Depressive Symptoms	Higher Level of Depressive Symptoms
	(n = 3488)	(n = 314)	(n = 2943)	(n = 918)
	n (%; 95% CI)	n (%; 95% CI)	n (%; 95% CI)	n (%; 95% CI)
**Activity context**				
*Train or compete with a sports club*				
≥1 time a week	1736 (51.6; 49.9–53.3)	111 (38.3; 32.7–43.9)	1473 (51.7; 49.9–53.6)	244 (38.5; 35.3–41.7)
<1 time a week	1631 (48.4; 46.8–50.1)	179 (61.7; 56.1–67.3)	1374 (48.3; 46.4–50.1)	549 (61.5; 58.3–64.7)
*Non-organised physical activity (football, snowboarding, or another type of physical activity)*				
≥1 time in the last 7 days	2098 (61.4; 59.8–63.1)	139 (45.6; 40.0–51.2)	1552 (53.5; 51.7–55.3)	375 (41.4; 38.2–44.6)
<1 time in the last 7 days	1318 (38.6; 37.0–40.2)	166 (54.4; 48.8–60.0)	1351 (46.5; 44.7–48.4)	531 (58.6; 55.4–61.8)
*Other leisure-time activities (associations, clubs, or societies)*				
≥3 times in the last month	1270 (38.6; 37.0–40.3)	140 (47.9; 42.2–53.7)	1199 (43.1; 41.3–45.0)	413 (47.6; 44.3–51.0)
<3 times in the last month	2016 (61.4; 59.7–63.0)	152 (52.1; 46.3–57.8)	1582 (56.9; 55.1–58.7)	454 (52.4; 49.0–55.7)
*Having friends*				
No	245 (7.1; 6.2–7.9)	66 (21.2; 16.6–25.7)	200 (6.8; 5.9–7.7)	150 (16.4; 14.0–18.1)
Yes	3216 (92.9; 92.1–93.8)	246 (78.8; 74.3–83.4)	2734 (93.2; 92.3–94.1)	764 (83.6; 81.2–86.0)
*Parents’ higher education*				
Parents’ education, both	1614 (56.9; 55.1–58.8)	128 (48.5; 42.5–54.5)	1503 (61.3; 59.4–63.2)	385 (52.2; 48.6–55.9)
Parents’ education, one of the parents	675 (23.8; 22.2–25.4)	64 (24.2; 19.1–29.4)	580 (23.7; 22.0–25.3)	205 (27.8; 24.6–31.1)
Parents’ education, neither of the parents	546 (19.3; 17.8–20.7)	72 (27.3; 21.9–32.7)	369 (15.0; 13.6–16.5)	147 (19.9; 17.1–22.8)
*Family economy*				
Good family economy	2762 (81.1; 79.8–82.4)	197 (64.0; 58.6–69.3)	2251 (78.3; 76.8–79.8)	536 (59.5; 56.3–62.7)
Neither good nor bad family economy	525 (15.4; 14.2–16.6)	68 (22; 17.5–26.7)	538 (18.7; 17.3–21.1)	266 (29.5; 26.5–32.5)
Poor family economy	118 (3.5; 2.9–4.1)	43 (14.0; 10.1–17.8)	86 (3.0; 2.4–3.6)	99 (11.0; 9.0–13.0)
*Do you smoke?*				
No	3343 (96.4; 95.8–97.0)	273 (87.8; 84.1–91.4)	2883 (98.6; 98.1–99.0)	845 (82.6; 90.9–94.3)
Yes	125 (3.6; 3.0–4.2)	38 (12.2; 8.6–15.9)	42 (1.4; 1.0–1.9)	68 (7.4; 5.7–9.2)
*Alcohol intoxication in the last 12 months*				
No intoxication episodes	3069 (88.9; 87.8–89.9)	223 (71.2; 66.2–76.3)	2666 (91.2; 90.2–92.2)	704 (76.9; 74.2–79.7)
Any intoxication episode	384 (11.1; 10.1–12.2)	90 (28.8; 23.7–33.8)	257 (8.8; 7.8–9.8)	211 (23.1; 20.3–25.8)

Higher level of depressive symptoms coded as ≥80th percentiles, lower level of depressive symptoms coded as <80th percentiles.

**Table 3 ijerph-19-10769-t003:** Youth data surveys: baseline characterstic of leisure time activities taking place in different contexts in adolescents aged 13–16 years in 2017–2019.

	None Organized PA	PA in a Sportsclub	Other Organized Activities
Variables			
** *Depressive symptoms* **			
Lower level	3536 (87.6%)	3289 (87.6%)	2518 (81.8%)
Higher level	500 (12.4%)	466 (12.4%)	560 (18.2%)
** *Having friends* **			
No	265 (6.2%)	259 (6.9%)	248 (8.0%)
Yes	4043 (93.8%)	3509 (93.1%)	2856 (92.0%)
** *Gender* **			
Male	2134 (53.7%)	1865 (50.6%)	1439 (47.0%)
Female	1839 (46.3%)	1822 (49.4%)	1623 (53.0%)
** *Parents higher education* **			
Both parents	2264 (61.7%)	2056 (64.0%)	1515 (58.2%)
One of the parents	847 (23.1%)	717 (22.3%)	630 (24.2%)
None of the parents	556 (15.2%)	438 (13.6%)	460 (17.7%)
** *Family economy* **			
Good economy	3468 (81.6%)	3071 (82.8%)	2320 (76.1%)
Neither good nor bad economy	657 (15.5%)	530 (14.3%)	582 (19.1%)
Poor economy	123 (2.9%)	110 (3.0%)	146 (4.8%)
** *Smoling* **			
No	4154 (96.7%)	3677 (97.7%)	2975 (96.3%)
Yes	142 (3.3%)	86 (2.3%)	115 (3.7%)
** *Alcohol intoxication last 12 month* **			
No intoxication episodes	3773 (88.1%)	3408 (90.8%)	2698 (87.3%)
Any intoxication episode	510 (11.9%)	346 (9.2%)	392 (12.7%)

Depressive symptoms coded as ≥80th percentiles, no depression symptoms coded as <80th percentiles. PA = Physical activity.

**Table 4 ijerph-19-10769-t004:** Binominal logistic regression of depressive symptoms in relation to different activity contexts.

	Crude Model	Adjusted Model	Crude Model	Adjusted Model	Crude Model	Adjusted Model
Variables	OR (95% CI)	OR (95% CI)	OR (95% CI)	OR (95% CI)	OR (95% CI)	OR (95% CI)
** *Activity context* **						
*Train or compete with a sports club*						
≥1 time a week	1 (ref)	1 (ref)				
<1 time a week	1.70 (1.50–1.93)	1.34 (1.15–1.57)				
*Non-organised physical activity (football, snowboarding, or another type of physical activity)*						
≥1 time in the last 7 days			1 (ref)	1 (ref)		
<1 time in the last 7 days			1.87 (1.65–2.11)	1.50 (1.29–1.74)		
*Other leisure-time activities (associations, clubs, or societies)*						
≥3 times in the last month					1 (ref)	1 (ref)
<3 times in the last month					0.76 (0.67–0.86)	0.79 (0.68–0.92)
*Having friends*						
Yes		1 (ref)		1 (ref)		1 (ref)
No		2.60 (2.08–3.25)		2.48 (1.99–3.10)		2.80 (2.24–3.50)
*Gender*						
Boy		1 (ref)		1 (ref)		1 (ref)
Girl		3.68 (3.12–4.35)		3.44 (2.92–4.05)		3.61 (3.05–4.26)
*Parents’ higher education*						
Parents’ education, both		1 (ref)		1 (ref)		1 (ref)
Parents’ education, one of the parents		1.10 (0.92–1.32)		1.10 (0.93–1.31)		1.13 (0.94–1.35)
Parents’ education, neither of the parents		1.10 (0.90–1.35)		1.11 (0.91–1.31)		1.17 (0.95–1.43)
*Family economy*						
Good family economy		1 (ref)		1 (ref)		1 (ref)
Neither good nor bad family economy		1.83 (1.53–2.18)		1.81 (1.52–2.15)		1.79 (1.50–2.15)
Poor family economy		3.30 (2.43–4.49)		3.28 (2.43–4.44)		3.60 (2.66–4.87)
*Do you smoke?*						
No		1 (ref)		1 (ref)		1 (ref)
Yes		2.82 (2.0–3.97)		2.68 (1.90–3.78)		2.70 (1.90–3.82)
*Alcohol intoxication in the last 12 months*						
No intoxication episodes		1 (ref)		1 (ref)		1 (ref)
Any intoxication episode		2.14 (1.75–2.63)		2.24 (1.83–2.74)		2.23 (1.82–2.75)

The dependent variable consisted of two categories coded as higher level of depressive symptoms (≥80 percentile) and lower level of depressive symptoms (<80 percentile). The reference category was lower level of depressive symptoms. Crude: bivariate analysis. OR: odds ratio.

**Table 5 ijerph-19-10769-t005:** Adjusted analysis of the moderation of family economy and physical activity in a sports club on the odds of depressive symptoms.

Interaction Variables	Depressive Symptoms	*p*-Value
	OR (95% CI)	
** *Good economy* **		
PA ≥ 1 time a week	1 (ref)	
PA < 1 time a week	1.56 (1.33–1.82)	<0.001
** *Neither good nor bad economy* **		
PA ≥ 1 time a week	1 (ref)	
PA < 1 time a week	1.60 (1.23–2.08)	0.001
** *Poor economy* **		
PA ≥ 1 time a week	1 (ref)	
PA < 1 time a week	0.86 (0.54–1.37)	0.54

Abbreviations: OR, odds ratio; PA, physical activity in a sports club. The outcome measure was defined as higher symptoms of depression (≥80 percentile) and lower symptoms of depression (<80 percentile). The reference category was lower symptoms of depression.

## Data Availability

The data and materials from the Ungdata Surveys are closed and stored in a national database administered by NOVA. The data are available for research purposes upon application. For request of the data, please contact ungdata@oslomet.no. Further information about the study and the questionnaires can be found on the web page (in Norwegian) (http://ungdata.no/, accessed on 15 January 2022).

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
