# Peer review of "Leisure-Time Activities in Different Contexts and Depressive Symptoms in Norwegian Adolescents: A Cross-Sectional Study"

_ijerph, 2022, doi:10.3390/ijerph191710769_

Round 1
Reviewer 1 Report
Thank you for opportunity to review an interesting topic „Leisure Time Activities in Different Context and Depressive Symptoms in Norwegian Adolescents: a Cross-Sectional Study“. The aim of present study was to examine the association between physical activity organized in sport clubs, non-organized physical activity, other organized leisure time activities, and depressive symptoms among adolescents.
However, I have minor concerns:
1. In the abstract, results, and conclusions, the authors interpret the results based on the magnitude of the odds ratios (ORs) (lower vs. higher). This is not correct from a statistical point of view. The magnitude of the odds ratios does not indicate the strength of the association between the variables have been analysed. This should be corrected. The ORs show only the association.
2. The introduction section seems should be complemented by a definition of depression and up-to-date data provided by the World Health Organisation (WHO).
3. Brief information on the prevalence/incidence of depressive symptoms in adolescent populations of other countries may be submitted to the introduction section.
4. I suggest that the authors could give an indication of possible reasons/potential factors why adolescents are at a higher-risk for depressive symptoms. There are plenty of literature on this topic.
5. The contents of the tables must be adapted in the correct English language. The current format is not well-organized.
6. Although the authors are discussing social contexts in the discussion, it is necessary to compare the results of the study with those specifically examining association between the physical activity and the severity of depressive symptoms.There is a lot of literature on this topic.
7. It would also be optimal to compare (discuss) the results of the study with those of randomised clinical trials that have (or have not) identified the effect of physical activity on the depression symptomatology.
8. It is also necessary to interpret the results in a more targeted way in the context of public health rather than in the social context.
9. In the conclusions section, it is best to generalise the results of this study. Practical recommendations and implications must focus on public health. I suggest that the authors adapt the conclusions in a more targeted way. There is a lack of specific implications related to the prevention of depressive symptoms and insufficient levels of physical activity. This is something that needs to be emphasized in this manuscript.
10. The back matter must be supplemented with the information associated with both Institutional Review Board Statement as well as Informed Consent Statement.
Kind Regards
Author Response
Reviewer 1 (R1):
(R1): Thank you for opportunity to review an interesting topic „Leisure Time Activities in Different Context and Depressive Symptoms in Norwegian Adolescents: a Cross-Sectional Study“. The aim of present study was to examine the association between physical activity organized in sport clubs, non-organized physical activity, other organized leisure time activities, and depressive symptoms among adolescents.
A: We highly appreciate your positive response to our paper. Thank you!
(R1): However, I have minor concerns:
- In the abstract, results, and conclusions, the authors interpret the results based on the magnitude of the odds ratios (ORs) (lower vs. higher). This is not correct from a statistical point of view. The magnitude of the odds ratios does not indicate the strength of the association between the variables have been analysed. This should be corrected. The ORs show only the association.
A: We agree that OR show the association between variables. In the present study the OR has been compared to the ref. category. Additionally, interaction analysis was used in the present study to examine the influence of gender and family economy on the strength of the relationship between physical activity in a sports club and depressive symptoms. Possible interaction effects were examined using LR-tests (Likelihood ratio test), contrasting models with and without interaction terms.
- The introduction section seems should be complemented by a definition of depression and up-to-date data provided by the World Health Organisation (WHO).
A: We agree and have added a definition of depression (page 2, line 29-31).
- Brief information on the prevalence/incidence of depressive symptoms in adolescent populations of other countries may be submitted to the introduction section.
A: We agree and have added this in the introduction (page 2, line 39-42).
- I suggest that the authors could give an indication of possible reasons/potential factors why adolescents are at a higher-risk for depressive symptoms. There are plenty of literature on this topic.
A: We agree and have added this in the introduction (page 2, line 32-34).
- The contents of the tables must be adapted in the correct English language. The current format is not well-organized.
A: We agree and have edited and changed the format in table 1, table 2 and table 4.
- Although the authors are discussing social contexts in the discussion, it is necessary to compare the results of the study with those specifically examining association between the physical activity and the severity of depressive symptoms. There is a lot of literature on this topic.
A: We agree and have added this in the discussion.
- It would also be optimal to compare (discuss) the results of the study with those of randomised clinical trials that have (or have not) identified the effect of physical activity on the depression symptomatology.
A: We agree and have added this in the discussion (page 15, line 406-413).
- It is also necessary to interpret the results in a more targeted way in the context of public health rather than in the social context.
A: We agree and have added this in the discussion (page 15, line 406-413).
- In the conclusions section, it is best to generalise the results of this study. Practical recommendations and implications must focus on public health. I suggest that the authors adapt the conclusions in a more targeted way. There is a lack of specific implications related to the prevention of depressive symptoms and insufficient levels of physical activity. This is something that needs to be emphasized in this manuscript.
A: We agree and have added this in the conclusion (page 18 (line 559-563).
- The back matter must be supplemented with the information associated with both Institutional Review Board Statement as well as Informed Consent Statement.
A: This has been added (page 18 and 19, line 570-579)
Reviewer 2 Report
Dear Authors,
First of all, I want to congratulate the authors on the conducted research and the obtained results.
I was very interested to be one of the first to review the obtained results and their interpretation. The conducted research is very large-scale and, in my opinion, allows us to better understand the factors associated with the mental health of the younger generation. It also has an important prognostic value and provides the necessary information for the construction and implementation of effective programs for young people, which are highly effective in terms of adjusting the level of physical activity and improving mental health, as well as promoting a healthy lifestyle.
It would be great if you could add more information about the validation of the tool for assessing depression symptoms, as well as the relevant link (line 206).
Best regards,
Author Response
Reviewer 2 (R2):
R2: First of all, I want to congratulate the authors on the conducted research and the obtained results. I was very interested to be one of the first to review the obtained results and their interpretation. The conducted research is very large-scale and, in my opinion, allows us to better understand the factors associated with the mental health of the younger generation. It also has an important prognostic value and provides the necessary information for the construction and implementation of effective programs for young people, which are highly effective in terms of adjusting the level of physical activity and improving mental health, as well as promoting a healthy lifestyle.
It would be great if you could add more information about the validation of the tool for assessing depression symptoms, as well as the relevant link (line 206).
A: Thank you for your valuable comments. We have added more information about the validation of the depressive symptoms scale (page 5, line 235-235), and also added a reference (line 223).
Reviewer 3 Report
Nice work with your manuscript. I liked how you framed your lit. Review. Thank you for your leisure time context definitions. I think that will help the readers. One suggestion you may want to think about is putting a little bit more emphasis on the importance of the family in aiding person. Overall, nice work.
Author Response
Reviewer 3 (R3):
R3:
Nice work with your manuscript. I liked how you framed your lit. Review. Thank you for your leisure time context definitions. I think that will help the readers. One suggestion you may want to think about is putting a little bit more emphasis on the importance of the family in aiding person. Overall, nice work.
A: We highly appreciate your positive response to our paper. We have added a sentence about the family in aiding person (page 18, line 562-564)